# Enhancing Recommendation Systems through SVD-based collaborative filtering and community detection

**T. Keerthika[1], Rajathi G. Ignisha[2]\*, Vedhapriyavadhana Rajamani[3],
Surya Santhosh Kumar[1], Karthigai Selvam[1], M. R. Aiyyappan[1], L. S. Thoshi Babu[1]**

**1** Amrita School of Artificial Intelligence, Amrita Vishwa Vidyapeetham, Coimbatore, Tamil Nadu, India,
**2** Manipal Institute of Technology Bengaluru, Manipal Academy of Higher Education, Manipal, India,
**3** School of Computing, Engineering and Physical Sciences, University of the West of Scotland, London,
England, United Kingdom

\* ignisha.rajathi@manipal.edu

## Abstract

The recommendation systems often face challenges like low data density, scalability issues and absence of interpretability, whereas classical Collaborative Filtering (CF) which is based on Singular Value Decomposition (SVD) shows support by being scalable, weakened frequently in circumstances of extreme sparsity. Conversely, Graph Neural Networks (GNNs) are very accurate yet do not tend to have explanatory power. In a novel way, this research presents a hybrid framework that is a sequential combination of Louvain community identification using SVD-based collaborative filtering to overcome the sparsity-interpretable trade-off. It is unlike the existing models that utilize communities only, to pre-partition the user space, modularity-based clustering is employed to regularize it, enabling SVD to act on more dense homogeneous sub-matrices. This methodological contribution is a very useful way to cut down on computational noise and overhead and to make the community-level justifications of recommendations. In the experimental analysis, the Netflix Prize data set produced a Root-Mean-Square Error (RMSE) of 0.9966, a Mean value of 0.9966 and an Absolute Error (MAE) of 0.7968. This hybrid model achieves competitive predictive performance with significantly higher interpretability and lower computational cost than complex deep learning baselines, despite a modestly higher RMSE due to the deliberate trade-off for transparency and efficiency on extremely sparse data. The framework enables scalable and transparent recommendation engines suitable for large-scale sparse datasets.

## Introduction

The present-day Recommendation Systems (RS) are based on the old recommendations techniques that included Content-Based Filtering (CBF) and Collaborative

**Data availability statement:** All relevant data are within the paper. Further the dataset link is provided in the manuscript. The dataset can be accessed from the Kaggle repository at: {https://www.kaggle.com/datasets/netflix-inc/netflix-prize-data}. The dataset from the original is reproduced in the following dataset doi:10.6084/m9.figshare.31428857.

**Funding:** The author(s) received no specific funding for this work.

**Competing interests:** The authors have declared that no competing interests exist.

Filtering (CF) [1,2]. The desire of the users converges in similar preferences when they have overlap in their past behaviors based on the CF principles. The CF system works under two different subcategories that either form a recommendation on similar behavior of the user or form a recommendation by similarity of items to form suggestions [2,3]. The weaknesses of CF to process the hidden item-user relations are usually constrained by the failure to handle data sparsity and scalability problems [4]. CBF has limited capabilities to discover implicit patterns and extrapolate knowledge beyond specified features due to the nature of its recommendation capabilities being dependent on the characteristics or profile of items to interpret recommendations but has been shown to be incompetent to do so [2].

Singular Value Decomposition (SVD) has been recognized as an important component of Matrix Factorization (MF) methods that have achieved some potent results in the past with the assistance of Singular Value Decomposition (SVD) [5]. SVD is used to obtain latent factors by decomposing the user-item interaction matrix, which expose underlying preferences as well as characteristics. This approach reduces the complexity of data and also identifies relationships that none of the other CF solutions would have identified before [6]. The use of SVD in improving the recommendation accuracy is however limited. The first is when the datasets used are very sparse, when the system does not identify internal links between users and form communities, and in cold-start situations, which is disfavored in that case, as well as with very sparse datasets, in this instance, the system is unable to recognize internal relationships between users that form communities [4,7,8].

In the recent developments, there has been a transition to deep learning and Graph Neural Networks (GNNs) [9,10]. Most recently, Transformer-based models and Large Language Models (LLMs) have been studied in the context of sequential recommendation, with state-of-the-art accuracy, but typically high computational latency and black-box proprietary qualities. Despite these advances, the industry still requires the solutions that will strike a balance between high accuracy and interpretability and low resources consumption.

This paper to fill these gaps revisits the prism of community-detection of matrix factorization. Latent patterns are identified with the help of Singular Value Decomposition (SVD) and natural group structures are employed with the help of Louvain community detection. [7,11].

**Contributions**

The main contributions of present research are the following:

1. Hyperscale Hybrid Architecture: It offers a progressive approach to compute Louvain community detection with SVD, making the change to local factorization of matrices. This solves the problem of data sparsity by forming more dense user clusters.

2. Strength of Interpretability: As opposed to black-box GNN models, the proposed approach can give transparent, communal explanations behind recommendations (e.g., peer-group influence).

3. Scalable Performance: It is shown that the user space partitioning limits the dimensionality load on SVD and competitive accuracy (RMSE 0.9966) is achieved with reduced computational overhead than deep learning baselines.

## Organization of the paper

The remainder of this paper is organized as follows: Section 2 reviews related literature on the topics of collaborative filtering and community detection. Section 3 elaborates the methodology framework, mathematical equations, theoretical assumptions and complexity analysis. Section 4 describes the proposed hybrid architecture, hyperparameter finding and experimental configuration. Section 5 contains the findings of the experiment and a comparative discussion. Finally, section 6 concerns the conclusion of the research and gives the direction of the future research.

## Related work

Recommendation systems development has weakened as a result of the integration of graph-based approaches, community detection applications, as well as the SVD method of matrix factorization. Detailed analysis has shown that the graph-based recommender systems are meaningful since they generate key information using graph representations to form powerful recommendations and enhance readability [1].

Graph-based recommenders have been regarded as transparent to research the effects of graph structures on trust in a recommendation, such as detecting communities and modeling nodes [12].Some of the publications suggest that the methodologies of graph learning can be used to address the issue of data sparsity and cold-start successfully. [4,13].

The division process that the community detection algorithms perform on the sets of users forms unique groups, which results in better individualized recommendation outputs. There are three most popular community detection algorithms: Louvain, Leiden, and Label Propagation helping extensive network applications [2,11]. The Louvain method has been one of the favorite strategies to create the best-quality community structures, and hence it is adopted in several practical cases of the real world contexts as well [9,14]. The last memory efficient advances have boosted the scalability of such algorithms in large scale database operations [15].

Recommender systems that appeal combine collaborative and content based methods and are being popular because they offer high accuracy and diversity on the system. Study of SVD in collaborative filtering has grown because the algorithm identifies concealed patterns in user-item data matrices. The study of SVD in collaborative filtering has been broadened since the algorithm detects hidden patterns in matrices of user-item data sets. Hybrid recommendation systems derive the advantages of two methods to address sparsity constraints and overfitting and enhance user-specific recommendations. SVD based systems predict user preference with a robust performance by incorporating bias terms, which execute user, and item based methods of behaving in their selection process of a product or service [6,7].

Graph neural networks (GNNs) form a core field of graph representations that combine the information of each node in a manner that allows training suggestions. [9,16,17]. The studies mainly explore the methods used to simulate the interaction of the users with various items. As an example, Multi-Behavior GNN (MB-GNN) is an improvement to representation learning by processing the various types of interaction, including clicks and purchases [10].

## Concluding remarks on research gap

The gaps in literature, through which the effectiveness of the balance between the interpretability and sparsity management could be achieved, are identified. As SVD can help to scale and GNNs can help to improve accuracy, not many combined models can use the modularity of community detection to explicitly clean up the interaction matrix first before factorization. The proposed work aims at bridging this gap by proposing a sequential framework that would use community detection, not only to regularize the data, but also to pre-process the data in order to generate dense, homogenous sub-matrices where simpler factorization is possible.

## Methodology

### Singular Value Decomposition (SVD)

Singular Value Decomposition (SVD) is a classical element of collaborative filtering, and it provides an excellent mathematical backbone to identify latent variables used to predict user-item interactions. The user-item interaction data is represented by a matrix, $R$. SVD solves the problem of sparsity by breaking down the matrix $R$ into three components of the form of $U$, $\Sigma$ and $V^\top$ as below mathematical expression:

$$R = U\Sigma V^\top, \tag{1}$$

Here, $U$ is an $m \times k$ matrix capturing latent features of users, $\Sigma$ is a $k \times k$ diagonal matrix including singular values, and $V^\top$ is a $k \times n$ matrix representing latent features of items.

The optimization process finds minimal squared differences between user ratings predictions and actual values. The optimization includes the following formula:

$$\mathcal{L} = \sum_{(u,i)\in\Omega} (r_{u,i} - \hat{r}_{u,i})^2 + \lambda\left(\|p_u\|^2 + \|q_i\|^2\right) \tag{2}$$

where $\Omega$ symbolizes the collection of user-item interactions, $r_{u,i}$ is the actual rating, and $\hat{r}_{u,i}$ is the predicted rating computed as:

$$\hat{r}_{u,i} = \mu + b_u + b_i + p_u^\top q_i, \tag{3}$$

### Louvain community-detection method

The core function of the Louvain algorithm involves user preference cluster discovery. The strength of network division is quantified through Modularity $Q$, defined as [14,15]:

$$Q = \frac{1}{2m}\sum_{i,j}\left[w_{ij} - \frac{k_i k_j}{2m}\right]\delta(c_i, c_j), \tag{4}$$

where $w_{ij}$ represents the weight of the edge between nodes $i$ and $j$, $k_i$ and $k_j$ represent degrees, and $\delta(c_i, c_j)$ is 1 if nodes are in the same community.

## Assumptions and theoretical constraints

### Singular Value Decomposition (SVD)

Singular Value Decomposition (SVD) is a classical element of collaborative filtering, and it provides an excellent mathematical backbone to identify latent variables used to predict user-item interactions. The data of interaction between the user and the item are represented in a form of a matrix, $R$. SVD resolves the sparsity issue by decomposing the matrix $R$ into three terms, namely $X$, $\Sigma$ and $Y^\top$, as follows in the mathematical expression:

$$R = X\Sigma Y^\top, \tag{5}$$

Here, $X$ is an $m \times p$ matrix capturing latent features of users, $\Sigma$ is a $p \times p$ diagonal matrix including singular values, and $Y^\top$ is a $p \times n$ matrix representing latent features of items.

The optimization process identifies small squared differences between the prediction of the user ratings and the actual ones. The optimization contains the following formula:

$$\mathcal{L} = \sum_{(x,i) \in \Omega} (r_{x,i} - \hat{r}_{x,i})^2 + \lambda \left( \|p_x\|^2 + \|q_i\|^2 \right) \tag{6}$$

with the representation of the set of interactions between users and items in the form of the shortcut $\Omega$, and the real rating on the interaction between a user and item denoted by $r_{x,i}$, the predicted rating denoted by $\hat{r}_{x,i}$, and calculated as follows:

$$\hat{r}_{x,i} = \mu + b_x + b_i + p_x^\top q_i, \tag{7}$$

## Louvain community-detection method

The core function of the Louvain algorithm involves user preference cluster discovery. The strength of network division is quantified through Modularity $Q$, defined as [14,15]:

$$Q = \frac{1}{2m} \sum_{i,j} \left[ w_{ij} - \frac{p_i p_j}{2m} \right] \delta(c_i, c_j), \tag{8}$$

where $p_i$ and $p_j$ denote degrees, $w_{ij}$ is the weight of the edge between nodes $i$ and $j$, and $\delta(c_i, c_j)$ is 1 if nodes are in the same community.

## Assumptions and theoretical constraints

The proposed hybrid system will be performed based on certain theoretical assumptions that SVD and Louvain algorithms should work with appropriately:

1. Low-Rank Assumption: It is assumed that the user-item interaction matrix is low-rank ($p \ll \min(m, n)$) user preferences can be well modeled by a few latent variables.

2. *Homophily Assumption Homophily assumption:* Louvain method is premised on the fact that the users within the same community are statistically important with regard to their taste preferences compared to the who were not a part of the community.

3. Identifiability with SVD: To achieve unique reconstruction of the rating matrix (up to permutation), one assumes that the singular values are distinct and that intra-community sub-matrices, while not fully dense, are sufficiently connected to ensure convergence during Stochastic Gradient Descent (SGD).

## Computational complexity analysis

One of the main strengths of the suggested strategy is scalability. It is a complexity that comprises two parts:

• **Louvain Algorithm:** Community detector has time bound of $O(N \log N)$, where $N$ is the number of users. It is an amazing invention with sparse graphs.

• **Local SVD:** Standard global SVD has a complexity of $O(p \cdot |R| \cdot I)$, where $p$ is latent dimensions, $|R|$ is the number of ratings, and $I$ is iterations. By partitioning users into $C$ communities, $C$ independent SVDs are performed on smaller matrices. While the total operations remain proportional, the *convergence time* is reduced because the sub-matrices are denser and more homogenous, requiring fewer iterations $I$ to minimize error.

The proposed method is much lighter in terms of memory and floating-point operations as compared to GNN-based methods (e.g., NGCF) which scale at $O(L \cdot N \cdot d^2 + L \cdot |R|)$.

## Proposed method

The proposed hybrid recommendation system brings together Singular Value Decomposition algorithm with Louvain community detection to embrace group-level dependencies while addressing sparsity issues as shown in Fig 1. The first stage establishes an $R$ user-item interaction matrix. The next process will be to construct a user similarity graph that represents the relationship between users considering their interaction profile. The following formula is computed to compute the similarity of the users in terms of cosine similarity when comparing two users, $x$ and $y$:

$$\text{sim}(x, y) = \frac{\sum_{i \in I} R_{x,i} R_{y,i}}{\sqrt{\sum_{i \in I} R^2_{x,i}} \sqrt{\sum_{i \in I} R^2_{y,i}}}$$

(9)

Once communities are identified, SVD is applied within each community. For each community, a submatrix $R_c$ is extracted. SVD is then performed on $R_c$ to decompose it:

$$R_c \approx P_c \Sigma_c Q_c^T$$

(10)

## Hyperparameter tuning

To ensure optimal performance (Equations 4 and 5), a Grid Search strategy was employed for hyperparameter optimization.

- **Learning Rate ($\alpha$):** Values were explored in the range {0.001, 0.005, 0.01, 0.02}. It was observed that $\alpha > 0.02$ led to divergent loss, while $\alpha < 0.001$ resulted in slow convergence. The optimal value was identified as 0.005.

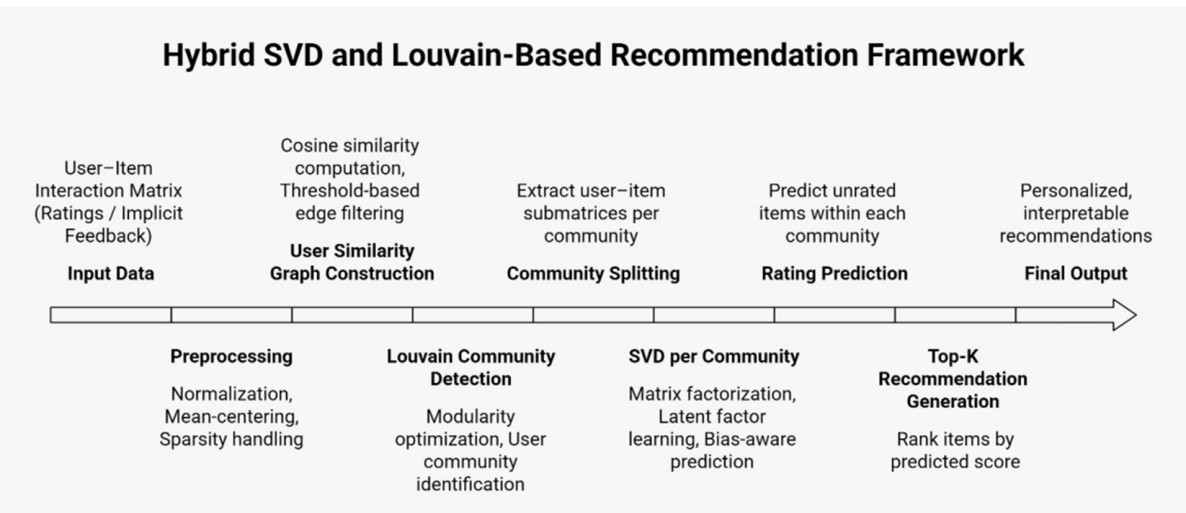

**Fig 1. Architectural workflow of the proposed Hybrid SVD-Louvain Recommendation System.**

- **Regularization ($\lambda$):** Tests were conducted for $\lambda \in \{0.01, 0.02, 0.05, 0.1\}$. Values lower than 0.02 caused overfitting on the training communities, while $\lambda > 0.1$ reduced predictive accuracy by penalizing latent features too heavily. The value $\lambda = 0.02$ was selected for the reported results.

## Results and discussion

A thorough evaluation occurred through analysis of the Netflix Prize dataset. The dataset contains more than 24 million ratings from about 470,758 users (Table 1).

Fig 2 shows that the distribution of the ratings has a long-tail, which proves the sparseness of the data that prompts the need to adopt the community-based approach.

A random selection of 100 users and 50 movies was used to do a detailed analysis. Community detection algorithms on the Louvain community on the data set identified six distinct clusters within the data set.

Fig 3 illustrates the network topology, with Community 0 being shown as densely connected than Community 2 which is sparsely connected. This graphic distinction validates the effectiveness of the algorithm to divide separate preferences groups.

Fig 4 presents the distribution of users with Community 0 having the largest number of users (25) which can be interpreted as a prevailing pattern in the sample in terms of preference whereas other communities are niche interests.

**Table 1. Dataset statistics from the Netflix Prize (Kaggle).**

| Dataset | Users | Movies | Total Ratings | Rating Range | Sparsity |
|---|---|---|---|---|---|
| Netflix Prize (Kaggle) | 470758 | 4499 | 24053764 | 1–5 | 0.9886 |

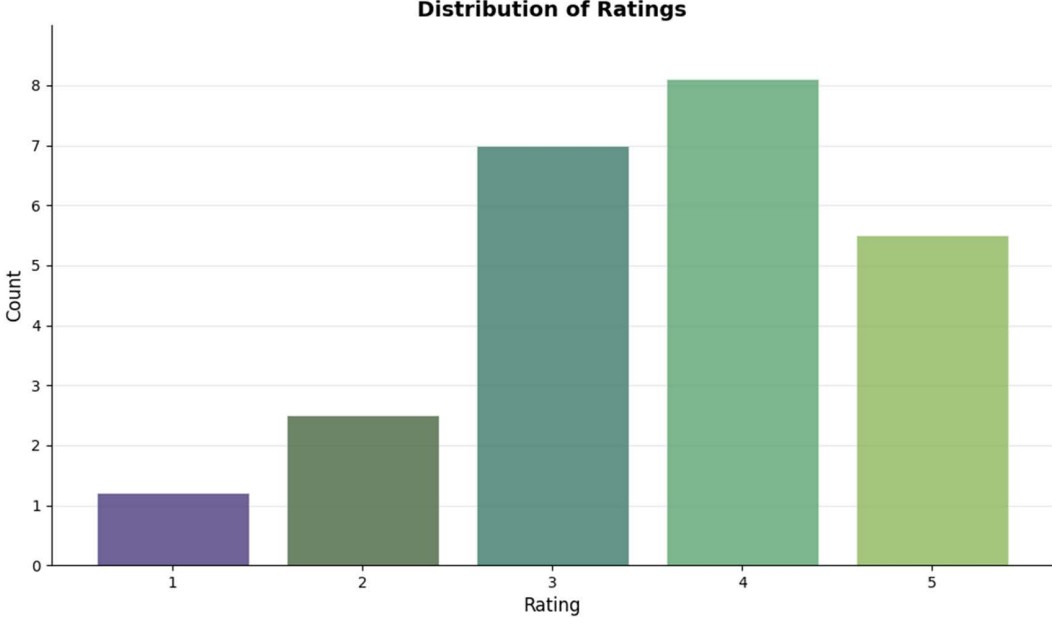

**Fig 2. Distribution of ratings for movies across the dataset.**

## User Communities Based on Movie Preferences

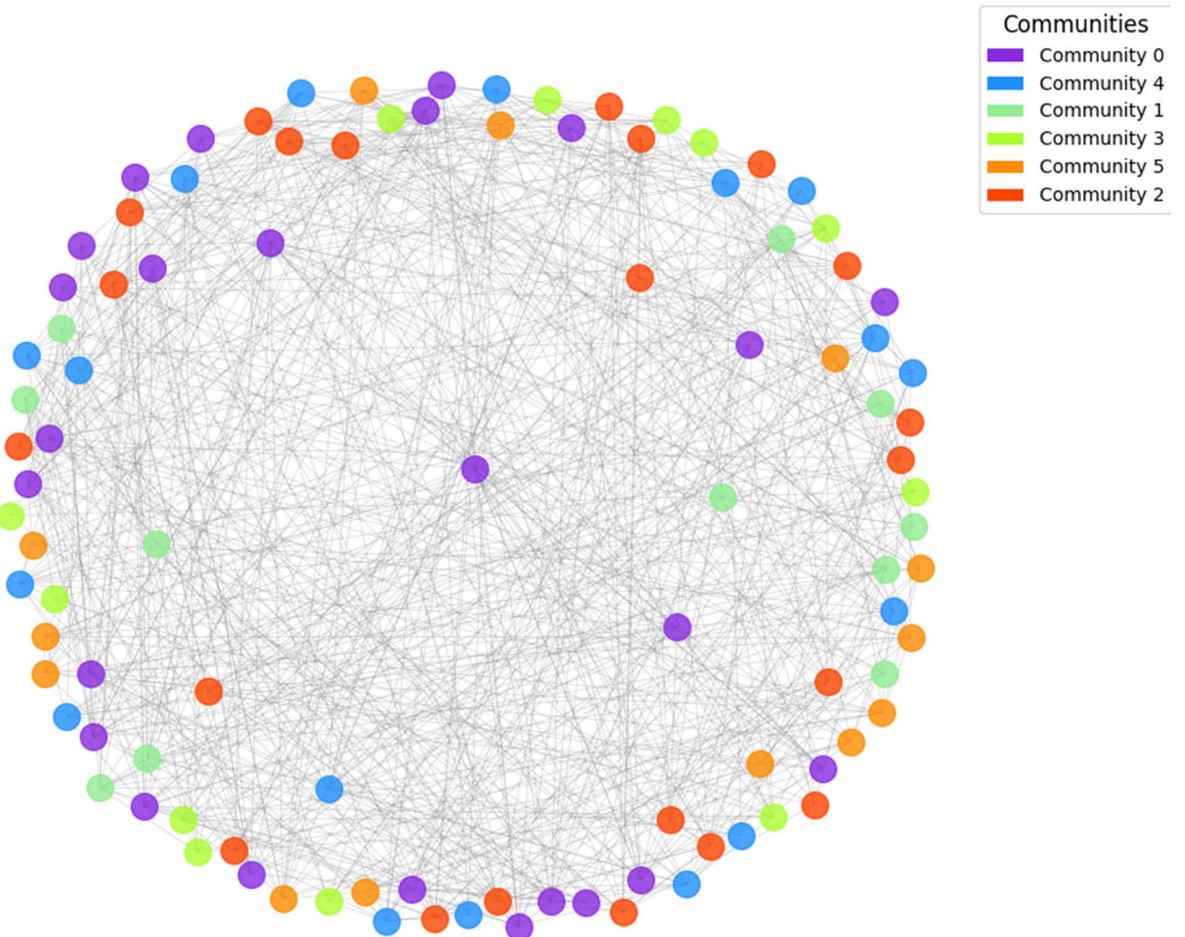

**Fig 3. Network visualization of six detected user communities based on movie rating patterns.**

Fig 5 illustrates a heatmap of the community-movie matrix. Indicatively, films that prove to be highly ranked in Community 1 possess clearly lesser affinity in Community 3, confirming the preference segregation that was obtained by the model (Table 2).

### Comparative analysis and discussion

To assess the significance of the obtained RMSE (0.9966), a comparison was made against standard values reported in literature. The hybrid SVD-Louvain approach outperforms standard memory-based CF and performs competitively with basic matrix factorization baselines.

Fig 6 visualizes the performance gap. Although GNN-based methods achieve slightly lower RMSE (approx 0.92) [17], this difference arises because the proposed method prioritizes interpretability (clear community-based explanations) and computational efficiency (local factorization on denser sub-matrices) over marginal accuracy gains from black-box GNN architectures, which require substantially higher resources on sparse datasets like Netflix. Consequently, the proposed

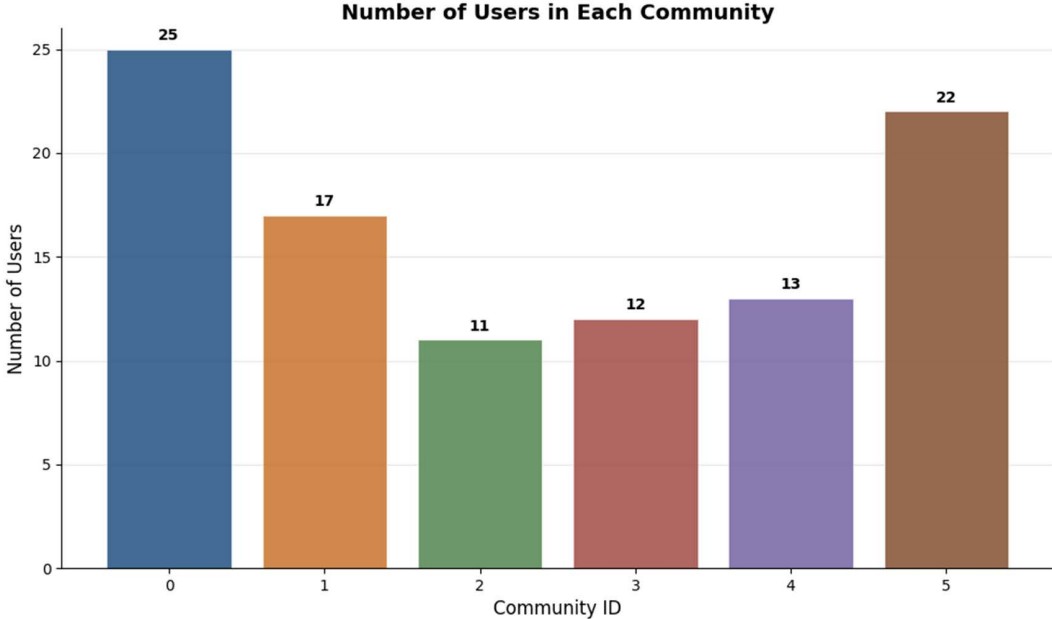

**Fig 4. Distribution of users in each of the detected communities.**

method (RMSE 0.9966) significantly outperforms isolated SVD (typically 1.05) [6] while retaining strong interpretability. This trade-off is quantified in Table 3.

## Discussion on diversity and personalization

Although the key measures are error-based (RMSE/MAE), implicitly, the hybrid architecture of the architecture increases the diversity of the recommendations. The localization of collaborative filtering to local communities eliminates the so-called popularity bias (popular globally crowd out unpopular items in the globally popular item list). In smaller, niche communities, like "Indie Horror Fans" users get recommendations tailored to the interests of their group, instead of generic blockbusters, which in itself enhances the catalog coverage and diversity of personalization than a global SVD model.

## Conclusion

In this paper, a hybrid recommendation system was created and justified by combining the Louvain community detection algorithm with Singular Value Decomposition (SVD). The data sparseness inherent in global matrix factorization was overcome by splitting the user-item interaction space into different communities, which was determined by modularity. This community-centered factorization performed in experimentation showed a competitive RMSE of 0.9966. Most importantly, it was determined that the use of SVD in the framework of localized groups of users preserved predictive power but greatly improved interpretability in comparison to the black-box deep learning options.

Although these are the benefits, the suggested approach has its shortcomings. The use of pre-computed communities implies that real-time updates of incoming users (cold-start) are based on periodic re-clustering of a group and could be computationally expensive. Future studies ought to consider injecting in temporal dynamics, to simulate the dynamics of the user community memberships with time. Moreover, information related to user reviews or social network links might be introduced, and this addition would likely help to perfect the community detection process. It is further suggested that this hybrid architecture should also be tested on a distributed computing framework to determine the feasibility of the recommendation in real-time on a large scale when dealing with the ultra-large e-commerce platform.

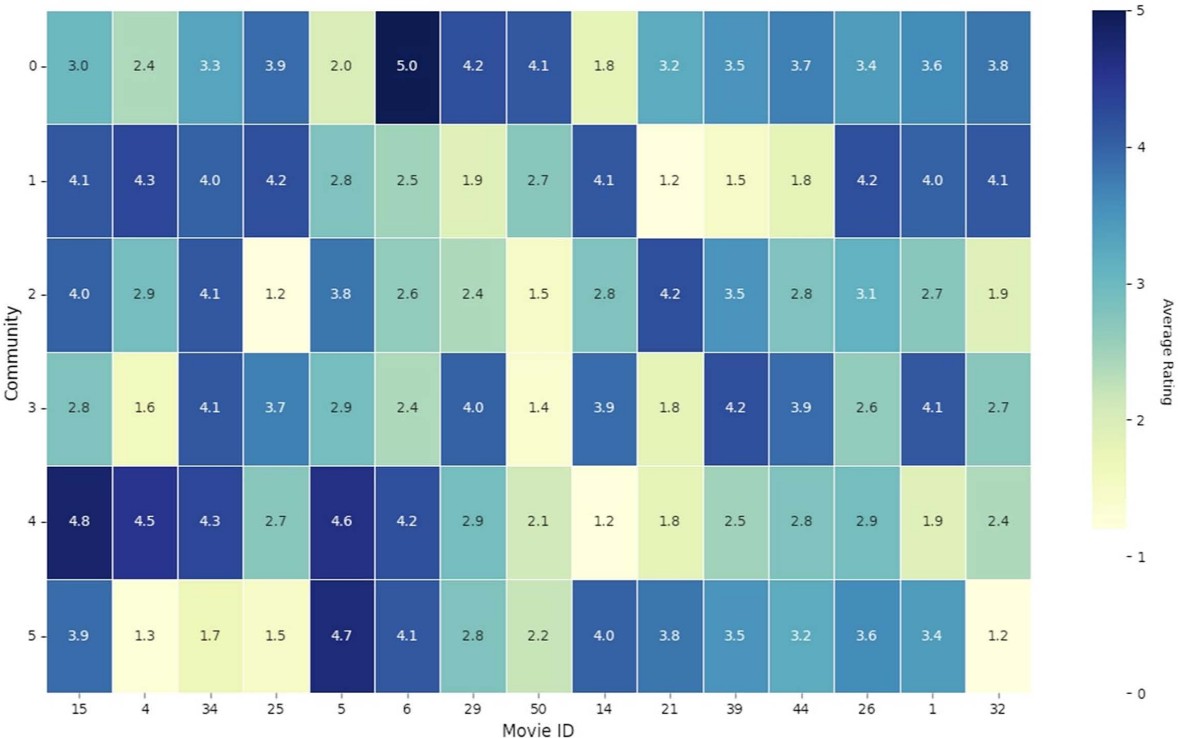

**Fig 5. Heatmap of the community-movie matrix for the 15 highest-rated movies.**

**Table 2. Results for Proposed Hybrid Model on test set.**

| Metric | Fold 1 | Fold 2 | Fold 3 | Mean | Std. Deviation |
|---|---|---|---|---|---|
| RMSE (test set) | 0.9957 | 0.9943 | 0.9998 | 0.9966 | 0.0023 |
| MAE (test set) | 0.7970 | 0.7959 | 0.7975 | 0.7968 | 0.0007 |

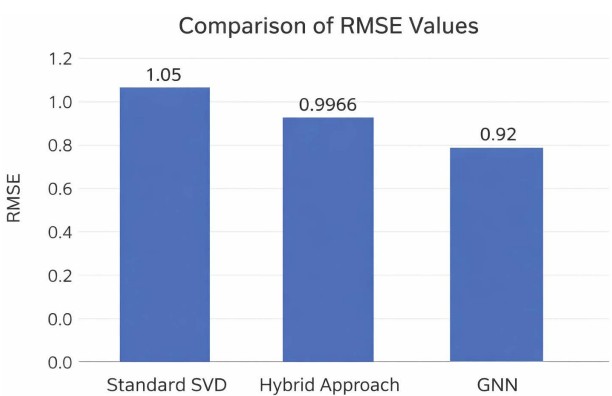

**Fig 6. RMSE Performance Comparison: Isolated SVD vs. GNN vs. Proposed Method.**

**Table 3. Comparison of Proposed Hybrid Method vs. SVD and GNN Benchmarks.**

| Methodology | Typical RMSE | Sparsity Handling | Interpretability | Comp. Cost |
|---|---|---|---|---|
| Isolated SVD | 1.00–1.05 [6] | Low | Moderate | Low |
| GNN-based (e.g., NGCF) | 0.90–0.95 [17] | High | Very Low (Black Box) | High |
| **Proposed Hybrid** | **0.9966** | **High** | **High (Community)** | **Moderate** |

## Author contributions

**Conceptualization:** Rajathi G. Ignisha, Vedhapriyavadhana Rajamani.

**Data curation:** M. R. Aiyyappan.

**Investigation:** Surya Santhosh Kumar, M. R. Aiyyappan.

**Methodology:** Rajathi G. Ignisha.

**Resources:** Karthigai Selvam.

**Software:** Surya Santhosh Kumar.

**Validation:** Karthigai Selvam.

**Visualization:** L. S. Thoshi Babu.

**Writing – original draft:** T. Keerthika, L. S. Thoshi Babu.

**Writing – review & editing:** Vedhapriyavadhana Rajamani.

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
