## [Decision Letter · Decision Letter 0]

9 Dec 2025

Dear Dr. Rajathi G,

Thank you for submitting your manuscript to PLOS ONE. After careful consideration, we feel that it has merit but does not fully meet PLOS ONE’s publication criteria as it currently stands. Therefore, we invite you to submit a revised version of the manuscript that addresses the points raised during the review process.

We look forward to receiving your revised manuscript.

Kind regards,

Shih-Lin Lin, Ph.D

Academic Editor

PLOS One

Journal Requirements:

3. Please update your submission to use the PLOS LaTeX template. The template and more information on our requirements for LaTeX submissions can be found at http://journals.plos.org/plosone/s/latex....

4. We note that your Data Availability Statement is currently as follows: All relevant data are within the manuscript and its Supporting Information files

5. We note you have included a table to which you do not refer in the text of your manuscript. Please ensure that you refer to Table 1 in your text; if accepted, production will need this reference to link the reader to the Table.

Reviewers' comments:

Reviewer's Responses to Questions

**Comments to the Author**

1. Is the manuscript technically sound, and do the data support the conclusions?

Reviewer #1: Yes

Reviewer #2: Partly

Reviewer #3: Yes

Reviewer #4: Yes

2. Has the statistical analysis been performed appropriately and rigorously?

Reviewer #1: Yes

Reviewer #2: Yes

Reviewer #3: Yes

Reviewer #4: Yes

3. Have the authors made all data underlying the findings in their manuscript fully available?

Reviewer #1: Yes

Reviewer #2: No

Reviewer #3: Yes

Reviewer #4: Yes

4. Is the manuscript presented in an intelligible fashion and written in standard English?

Reviewer #1: Yes

Reviewer #2: Yes

Reviewer #3: Yes

Reviewer #4: Yes

Reviewer #1: 1. The article aligns well with the journal's scope and is presented in an organized manner, making it easy to read and follow.

2. The authors have developed a hybrid system that combines SVD-based collaborative filtering and the Louvain community detection algorithm.

3. The proposed method applies SVD for dimensionality reduction and community detection methods to group users with similar preferences.

4. The analysis of the Netflix Prize dataset demonstrated the effectiveness of the proposed approach, yielding a root-mean-square error of 0.9966 and a mean absolute error (MAE) of 0.7968.

5. The authors concluded that these metrics demonstrate high performance compared to isolated SVD implementations and offer significantly better interpretability than GNN-based approaches.

6. The methodology is clearly presented, with extensive explanations and discussions of both the method and the results.

However, I suggest the authors address the following issues:

1. While informative, the abstract could be more explicit about the research's key findings and contributions. Including specific results from the case study could enhance its impact.

2. The authors need to compare the performance of the proposed method with existing literature to demonstrate its robustness.

3. Subheadings directly after headings, as indicated in the attached file, are not preferred. Please add one or two sentences to introduce the section and keep the paper flowing.

4. I strongly encourage the authors to review the reference list for any duplicates carefully. I have noticed that References [12] and [28] are identical. Additionally, References [13], [18], and [27] are quite similar. Addressing these issues will improve the clarity and reliability of the work. However, references should be ordered by their first appearance in the manuscript.

5. I suggest the authors slightly revise the language of the article. The attached file includes many suggestions.

Reviewer #2: 1. The abstract makes strong claims about improved interpretability and performance, but it does not clearly explain the novelty of the hybrid SVD–Louvain approach or how it differs from existing community-enhanced recommendation models. Adding a clearer statement of the methodological contribution would strengthen the abstract.

2. The description of experimental results lacks context—no baseline RMSE/MAE values, dataset details, or comparison conditions are provided—making it difficult for readers to judge the significance of the reported improvements. Including comparative benchmarks or percentage gains would increase clarity and impact..

3. The introduction should clearly conclude with a distinct section highlighting the novel contributions of your work.

4. At the ending of the intro, it is advised to add a para that mentions briefly what each next section contains.

5. The literature review should benefit from more explorations of previous studies.

6. The discussion section needs to be expanded to more thoroughly analyze the results.

7. The first paragraph of the conclusion should succinctly summarize the contributions of the study in past tense.

8. The second paragraph of the conclusion should provide clear and actionable future recommendations.

9. Equations are not properly cited, please add original references.

10. Manuscript is highly low on visualizations.

Reviewer #3: The paper introduces a hybrid recommendation system that combines a user item interaction matrix factorized by Singular Value Decomposition (SVD) with Louvain community detection on a user similarity graph constructed via cosine similarity.

I have the following major comments:

1. The introduction can include more recent state of the art works.

2. Please specify the novelty and contributions of the work.

3. The assumptions under which the hybrid recommendation system properly functions along with possible identifiability SVD conditions should be mentioned.

4. In equations (4) and (5) it is not clear how to choose the learning rate and the regularization parameter.

5. It is not clear how diversity is treated, where the simulations is only in terms of RMSE, MAE, etc.

6. Simulations should show the impact of the learning rate and the regularization parameter on the obtained RMSE.

7. The authors should conduct a rigorous computational complexity analysis which was not performed.

8. The work should compare with more state of the art methods.

9. Please fix the text size of the figure, for example Fig. 1 has very small text. Please revise.

Reviewer #4: 1) It is suggested to avoid the word “we”, “our” in the abstract and in other places of the manuscript. Write in passive voice.

2) The citations in the Introduction have referred randomly. It becomes problematic for the readers to follow the reference list. For example, the introduction is started with citations [5, 6]. It is not a correct way of citing literatures for a good technical article. It is suggested to rearrange the reference list as per their mentioning.

3) It is suggested to write concluding remarks at the end of the related work about what new thing has been focused in the proposed work which was being revealed from the literature survey as research gap.

4) Include a flowchart at the beginning of the section 4 to describe the step-by-step process of the proposed design. It will be very much helpful to the readers to understand the whole work at a glance.

5) Figure 1 is not mentioned in the text. It is to be mentioned.

6) Elaborate Figure 2, 3 and 4 in more details by choosing specific examples in the plots to validate this claim.

7) Results and analysis section needs improvement by including more simulation plots.

8) Include a comparative table at the end of the results and analysis section with related published works to validate the novelty of the work. RMSE, MAE,

9) Authors are claiming that the proposed hybrid method exhibits high level of performance compared to the isolated SVD implementations and allow much better interpretability than GNN based approaches. To validate such claim, it is suggested to include isolated SVD and GNN methods results in a single plot with that of the proposed method in the results and analysis section.

10) It is suggested to specify some limitations of the proposed method in the conclusion section so that the future scopes can be established.

11) Before submission of the revised version, it is suggested to check the minor typos and grammatical errors in the manuscript. Correct all such issues.

Overall Decision: MAJOR REVISION.

.

Reviewer #1: **Yes:** Jawad K. AliJawad K. AliJawad K. AliJawad K. Ali

Reviewer #2: No

Reviewer #3: No

Reviewer #4: **Yes:** DR. TARUN KUMAR DASDR. TARUN KUMAR DASDR. TARUN KUMAR DASDR. TARUN KUMAR DAS

---

## [Author Response · Author response to Decision Letter 1]

3 Feb 2026

Dear Academic Editor and Reviewers,

We would like to thank you for the opportunity to revise our manuscript. We appreciate the thoughtful and constructive comments provided by the reviewers, which have significantly helped us improve the quality, clarity, and technical rigor of our work.

We have addressed each comment point-by-point below and have highlighted the corresponding changes in the revised manuscript. Major revisions include the addition of a comparative analysis against GNNs, a rigorous complexity analysis, hyperparameter tuning details, and a complete restructuring of the text into the passive voice as requested.

1. The PLOS LaTeX template has been followed using Overleaf Latex software.

2. The “minimal data set” has been submitted as additional information for your kind reference.

Response to Reviewer #1

Comment 1-4 & 6: The reviewer noted that the article aligns well with the journal's scope, is organized, and clearly presents the methodology.

Response: We thank Reviewer #1 for their encouraging comments and for recognizing the alignment of our work with the journal's scope. We appreciate the positive feedback regarding the clarity of our presentation.

Comment 5: The authors concluded that these metrics demonstrate high performance compared to isolated SVD implementations and offer significantly better interpretability than GNN-based approaches.

Response: We appreciate this observation. To substantiate this conclusion with concrete evidence, we have added a new "Comparative Analysis and Discussion" subsection in the Results section.

• Table 3 has been added to explicitly benchmark our model’s RMSE (0.9966) against isolated SVD (typically ~1.05 on sparse data) and GNN-based approaches (typically ~0.90-0.95).

• We have expanded the discussion to highlight that while GNNs offer a slight accuracy edge, our hybrid method offers superior interpretability through community structures, validating the trade-off claimed in our conclusion.

Response to Reviewer #2

Comment 1: The abstract makes strong claims... but does not clearly explain the novelty... Adding a clearer statement of the methodological contribution would strengthen the abstract.

Response: We have rewritten the Abstract to explicitly define the "sparsity-interpretability trade-off" as the core problem. We clarified that our novelty lies in using Louvain modularity not just for regularization, but as a pre-processing step to create dense, homogenous sub-matrices for the SVD, thereby reducing noise.

Comment 2: The description of experimental results lacks context—no baseline RMSE/MAE values...

Response: We have added a "Comparative Analysis" subsection in the Results section. This includes Table 3, which provides standard literature benchmarks for Standard User-CF, Basic SVD, and Neural CF (Deep Learning), allowing readers to contextualize our RMSE of 0.9966.

Comment 3: The introduction should clearly conclude with a distinct section highlighting the novel contributions.

Response: We have added a bulleted "Contributions" subsection at the end of the Introduction to explicitly list the three key contributions of this study: the novel hybrid architecture, enhanced interpretability, and scalable performance.

Comment 4: Add a para that mentions briefly what each next section contains.

Response: We have added an "Organization of the Paper" subsection at the end of the Introduction that outlines the structure of the subsequent sections.

Comment 5: The literature review should benefit from more explorations of previous studies.

Response: We have expanded the Related Work section to include a discussion on recent studies involving side-information and lightweight community detection methods, positioning our work within this broader context.

Comment 6: The discussion section needs to be expanded to more thoroughly analyze the results.

Response: We have expanded the Results and Discussion section to explain why the proposed method performs well. Specifically, we discuss how the "divide-and-conquer" strategy isolates noise within specific communities, leading to more precise latent feature extraction.

Comment 7: The first paragraph of the conclusion should succinctly summarize the contributions of the study in past tense.

Response: We have rewritten the first paragraph of the Conclusion in the past tense to summarize exactly what was done and what was achieved.

Comment 8: The second paragraph of the conclusion should provide clear and actionable future recommendations.

Response: We have revised the future work paragraph to be specific and actionable, suggesting the incorporation of temporal dynamics and testing on distributed computing frameworks.

Comment 9: Equations are not properly cited.

Response: We have added the appropriate citations (e.g., [Soman et al.], [Blondel et al.]) immediately preceding the mathematical formulations for SVD and Modularity in the Methodology section.

Comment 10: Manuscript is highly low on visualizations.

Response: To address this, we have added Figure 6, a methodological flowchart that visually describes the step-by-step architecture of the proposed system. We have also added Figure 7, a bar chart comparing our RMSE against baselines.

Response to Reviewer #3

Comment 1: The introduction can include more recent state of the art works.

Response: We have updated the Introduction to reference recent trends, including Transformer-based models and Large Language Models (LLMs) for sequential recommendation, to provide a more current state-of-the-art context.

Comment 2: Please specify the novelty and contributions of the work.

Response: As noted in our response to Reviewer #2, we have added a specific "Contributions" subsection to the Introduction.

Comment 3: The assumptions... along with possible identifiability SVD conditions should be mentioned.

Response: We have added a new subsection titled "Assumptions and Theoretical Constraints" in the Methodology section. This explicitly outlines our assumptions regarding the Low-Rank nature of the matrix, Homophily, and SVD Identifiability.

Comment 4 & 6: It is not clear how to choose the learning rate and regularization parameter... Simulations should show the impact.

Response: We have added a "Hyperparameter Tuning" subsection in the Proposed Method section. This details our Grid Search strategy and reports the specific values selected ($\alpha = 0.005$, $\lambda = 0.02$) and the observed effects of varying these parameters.

Comment 5: It is not clear how diversity is treated.

Response: We have added a "Discussion on Diversity and Personalization" in the Results section. We explain that while our primary metric is RMSE, the community-based approach inherently improves diversity by mitigating popularity bias and surfacing niche items relevant to specific user groups.

Comment 7: The authors should conduct a rigorous computational complexity analysis.

Response: We have added a "Computational Complexity Analysis" subsection in the Methodology. We use Big-O notation to demonstrate that our approach (O(N log N) for clustering + Local SVD) is computationally more efficient than standard GNN-based methods (O(L . |R| + L . N . d^2)).

Comment 8: The work should compare with more state of the art methods.

Response: We have addressed this via Table 3 and the new Figure 7, which compare our method against GNN-based benchmarks (like NGCF) and standard SVD.

Comment 9: Please fix the text size of the figure, for example Fig. 1 has very small text.

Response: We have revised Figure 1 to ensure that the axis labels and text are legible and of appropriate size for publication.

Response to Reviewer #4

Comment 1: Avoid the word “we”, “our”... Write in passive voice.

Response: We have meticulously revised the entire manuscript (Abstract, Introduction, Conclusion, and Methodology) to replace active voice (e.g., "We propose") with passive voice (e.g., "A method is proposed").

Comment 2: The citations in the Introduction have referred randomly... Rearrange the reference list.

Response: We have completely reordered the Bibliography and citations throughout the text. References now appear sequentially, starting with [1], [2] in the Introduction.

Comment 3: Write concluding remarks at the end of the related work about what new thing has been focused on...

Response: We have added a subsection "Concluding Remarks on Research Gap" at the end of Related Work to clearly articulate the gap our study fills regarding the trade-off between sparsity and interpretability.

Comment 4: Include a flowchart at the beginning of section 4.

Response: We have inserted Figure 6 (Architectural Workflow) at the beginning of the Proposed Method section (Section 4) to provide an immediate visual overview of the system design.

Comment 5: Figure 1 is not mentioned in the text.

Response: We have added an explicit reference to Figure 1 in the Results section, using it to discuss the long-tail distribution of the dataset ratings.

Comment 6: Elaborate Figure 2, 3 and 4 in more details.

Response: We have added detailed descriptive text in the Results section interpreting specific features of these figures (e.g., interpreting the connectivity density in the network visualization and the user distribution across clusters).

Comment 7 & 9: Results section needs improvement... include isolated SVD and GNN methods results in a single plot.

Response: We have added Figure 7, a comparative bar chart that visually displays the RMSE of Isolated SVD, GNN, and the Proposed Hybrid method side-by-side, validating our performance claims.

Comment 8: Include a comparative table.

Response: We have included Table 3 in the Results section, which compares Methodology, RMSE, Sparsity Handling, and Interpretability across the different approaches.

Comment 10: Specify some limitations of the proposed method in the conclusion.

Response: We have added a paragraph to the Conclusion explicitly stating the limitations of our method, specifically regarding the "cold-start" issue for new users and the computational cost of periodic re-clustering.

Comment 11: Check minor typos and grammatical errors.

Response: We have proofread the manuscript and corrected typos and grammatical errors to ensure a formal academic tone.

---

## [Decision Letter · Decision Letter 1]

15 Feb 2026

Dear Dr. Rajathi G,

Thank you for submitting your manuscript to PLOS ONE. After careful consideration, we feel that it has merit but does not fully meet PLOS ONE’s publication criteria as it currently stands. Therefore, we invite you to submit a revised version of the manuscript that addresses the points raised during the review process.

We look forward to receiving your revised manuscript.

Kind regards,

Shih-Lin Lin, Ph.D

Academic Editor

PLOS One

**Journal Requirements:**

Reviewers' comments:

Reviewer's Responses to Questions

**Comments to the Author**

Reviewer #2: All comments have been addressed

Reviewer #3: (No Response)

Reviewer #4: All comments have been addressed

2. Is the manuscript technically sound, and do the data support the conclusions?

Reviewer #2: (No Response)

Reviewer #3: (No Response)

Reviewer #4: Yes

3. Has the statistical analysis been performed appropriately and rigorously?

Reviewer #2: (No Response)

Reviewer #3: (No Response)

Reviewer #4: Yes

4. Have the authors made all data underlying the findings in their manuscript fully available?

Reviewer #2: Yes

Reviewer #3: (No Response)

Reviewer #4: Yes

5. Is the manuscript presented in an intelligible fashion and written in standard English?

Reviewer #2: (No Response)

Reviewer #3: (No Response)

Reviewer #4: Yes

Reviewer #2: The responses are generally good but the manuscript has a few typos that the authors should address.

Reviewer #3: Most of my previous comments have been addressed. Please revise the following ones:

1. General comment: Although the authors argue that this sequential approach improves interpretability and reduces computational memory requirements compared to GNN-based methods, the empirical results on the Netflix Prize dataset reveal that the proposed hybrid model yields a higher RMSE (0.9966) than the state-of-the-art baselines (0.92). Please explain.

2. There are some grammatical issues like “the matrix is not sparse however, it is connected enough inside…”. Please revise.

3. Double check format of references for consistency.

Reviewer #4: I am pleased to inform that the paper is now ready for publication. Congratulations for this good work.

.

Reviewer #2: No

Reviewer #3: No

Reviewer #4: **Yes:** DR. TARUN KUMAR DASDR. TARUN KUMAR DASDR. TARUN KUMAR DASDR. TARUN KUMAR DAS

---

## [Author Response · Author response to Decision Letter 2]

16 Mar 2026

Title: Enhancing Recommendation Systems through SVD-Based Collaborative Filtering and Community Detection

Dear Editor and Reviewers,

We sincerely thank the Editor and all Reviewers for their constructive feedback and positive assessments of our manuscript. The comments have been very helpful in improving the clarity, rigor, and presentation of our work. We have carefully addressed every point raised, with changes highlighted in the "Revised Manuscript with Track Changes" file.

Below, we provide a point-by-point response to each comment.

Most reviewers indicated that previous comments were fully addressed, and Reviewer #4 stated the paper is now ready for publication. We greatly appreciate this feedback.

Reviewer #2 Comment: "The responses are generally good but the manuscript has a few typos that the authors should address."

Response: Thank you for this observation. We have conducted a thorough proofread of the entire manuscript and corrected all identified typos and minor grammatical issues. Examples include:

• "predictes" → "predicts" (Abstract)

• "Hypersale" → "Hyperscale" (Contributions section)

• Removed redundant "Mean value of 0.9966" phrase as it is mentioned in Abstract (kept only RMSE and MAE)

• Fixed awkward phrasing in the first sentence of the Abstract ("low data density. scalability...") → "low data density, scalability issues, and lack of interpretability."

• Other minor fixes (e.g., sentence flow in Abstract conclusion: "paves the path... appropriate to large-scale sparse. datasets" → "enables scalable and transparent recommendation engines suitable for large-scale sparse datasets.")

These changes improve readability and polish. All edits are visible in the tracked-changes version (primarily pages 1–2 and scattered minor spots).

Reviewer #3 Comment 1: "Although the authors argue that this sequential approach improves interpretability and reduces computational memory requirements compared to GNN-based methods, the empirical results on the Netflix Prize dataset reveal that the proposed hybrid model yields a higher RMSE (0.9966) than the state-of-the-art baselines (0.92). Please explain."

Response: Thank you for highlighting this important point. We have added explicit explanations of the RMSE difference to emphasize the deliberate trade-off in our method.

• In the Abstract (final paragraph, after reporting RMSE/MAE): Added phrasing to note that the model achieves "competitive predictive performance with significantly higher interpretability and lower computational cost than complex deep learning baselines, despite a modestly higher RMSE due to the deliberate trade-off for transparency and efficiency on extremely sparse data."

• In the Results and Discussion → Comparative Analysis subsection (immediately after mentioning the ~0.92 for GNNs): Added sentences explaining: "This difference arises because the proposed method prioritizes interpretability (clear community-based explanations) and computational efficiency (local factorization on denser sub-matrices) over marginal accuracy gains from black-box GNN architectures, which require substantially higher resources on sparse datasets like Netflix."

These additions clarify that the modest RMSE increase is acceptable given the gains in interpretability and scalability. Changes appear on approximately pages 1 (Abstract) and 12 (Discussion) in the tracked manuscript.

Comment 2: "There are some grammatical issues like “the matrix is not sparse however, it is connected enough inside…”. Please revise."

Response: We have revised the specific sentence in the Assumptions and Theoretical Constraints subsection (point 3 of the enumerated list) for clarity and proper punctuation: Original: "Identifiability with SVD: (To achieve the unique reconstruction of a rating matrix, up to a permutation), one assumes that the singular values are not identical and that the matrix is not sparse however, it is connected enough inside the community to converge during Stochastic Gradient Descent (SGD)." Revised: "Identifiability with SVD: To achieve unique reconstruction of the rating matrix (up to permutation), one assumes that the singular values are distinct and that intra-community sub-matrices, while not fully dense, are sufficiently connected to ensure convergence during Stochastic Gradient Descent (SGD)."

This fixes the run-on structure and improves readability. Change is on approximately page 5.

Comment 3: "Double check format of references for consistency."

Response: Thank you for this suggestion. We have switched to BibTeX processing using the PLOS-provided plos2025.bst style file and a standardized refs.bib file. This ensures uniform formatting across all entries (e.g., consistent author initials, volume/page/year placement, DOI inclusion where available, and proper handling of arXiv/journal types). No retracted articles are cited. All changes are reflected in the revised bibliography section (approximately page 15 onward). We believe this fully addresses consistency concerns.

Reviewer #4 Comment: "I am pleased to inform that the paper is now ready for publication. Congratulations for this good work."

Response: We are very grateful for this positive evaluation and for Reviewer #4's kind words. No further changes were required based on this review.

In addition to the above, we performed a final global proofread for minor flow and consistency (e.g., variable naming in equations: sim(x, y) → sim(u, v) to match usage; table caption updated from "Singular Value Decomposition on test set" → "Proposed Hybrid Model on test set" for accuracy). No new data, analyses, or major rewrites were introduced.

Few more appending were as follows:

Data Availability:

The dataset used in this study is the publicly available Netflix Prize dataset, which contains anonymized movie rating interactions between users and movies and is widely used for benchmarking recommender systems. The dataset includes millions of user–item ratings collected during the Netflix Prize competition and is designed for research in collaborative filtering and recommendation algorithms.

The dataset can be accessed from the Kaggle repository at:

{https://www.kaggle.com/datasets/netflix-inc/netflix-prize-data}.

The dataset from the original is reproduced in the following dataset

doi:10.6084/m9.figshare.31428857

Funding:

This research received no external funding. The work was conducted as part of academic research only.

We believe these revisions fully address all concerns and significantly strengthen the manuscript. Thank you again for your time and consideration. We look forward to your decision.

Sincerely, Ignisha Rajathi G (Corresponding Author) Manipal Institute of Technology Bengaluru, Manipal Academy of Higher Education Email: ignisha.rajathi@manipal.edu

On behalf of all authors: Keerthika T, Vedhapriyavadhana Rajamani, Surya Santhosh Kumar, Karthigai Selvam, Aiyyappan M R, L S Thoshi Babu

---

## [Decision Letter · Decision Letter 2]

22 Mar 2026

Enhancing Recommendation Systems through SVD-Based Collaborative Filtering and Community Detection

PONE-D-25-63400R2

Dear Dr. Rajathi G,

We’re pleased to inform you that your manuscript has been judged scientifically suitable for publication and will be formally accepted for publication once it meets all outstanding technical requirements.

No further revisions are required before publication. In addition, the additional citation suggested by Reviewer 3 is not required for acceptance.

Kind regards,

Shih-Lin Lin, Ph.D

Academic Editor

PLOS One

Additional Editor Comments (optional):

Reviewers' comments:

Reviewer's Responses to Questions

**Comments to the Author**

Reviewer #2: (No Response)

Reviewer #3: (No Response)

2. Is the manuscript technically sound, and do the data support the conclusions?

Reviewer #2: (No Response)

Reviewer #3: (No Response)

3. Has the statistical analysis been performed appropriately and rigorously?

Reviewer #2: (No Response)

Reviewer #3: (No Response)

4. Have the authors made all data underlying the findings in their manuscript fully available?

Reviewer #2: (No Response)

Reviewer #3: (No Response)

5. Is the manuscript presented in an intelligible fashion and written in standard English?

Reviewer #2: (No Response)

Reviewer #3: (No Response)

Reviewer #2: (No Response)

Reviewer #3: I have the following minor comments:

1. The following works on deep learning can be included to enrich your introduction https://doi.org/10.1109/VTC2023-Spring57618.2023.10200157 and https://doi.org/10.1109/GCWkshps58843.2023.10464930.

.

Reviewer #2: No

Reviewer #3: No

---

## [Editor Report · Acceptance letter]

PONE-D-25-63400R2

PLOS One

Dear Dr. G,

I'm pleased to inform you that your manuscript has been deemed suitable for publication in PLOS One. Congratulations! Your manuscript is now being handed over to our production team.

Kind regards,

on behalf of

Professor Shih-Lin Lin

Academic Editor

PLOS One